# Generalization bounds
# for deep convolutional neural networks

**Philip M. Long**[*] **and Hanie Sedghi**[*]
Google Brain
{plong,hsedghi}@google.com

## Abstract

We prove bounds on the generalization error of convolutional networks. The bounds are in terms of the training loss, the number of parameters, the Lipschitz constant of the loss and the distance from the weights to the initial weights. They are independent of the number of pixels in the input, and the height and width of hidden feature maps. We present experiments using CIFAR-10 with varying hyperparameters of a deep convolutional network, comparing our bounds with practical generalization gaps.

## 1 Introduction

Recently, substantial progress has been made regarding theoretical analysis of the generalization of deep learning models (see Neyshabur et al., 2015; Zhang et al., 2016; Dziugaite and Roy, 2017; Bartlett et al., 2017; Neyshabur et al., 2017; 2018; Arora et al., 2018; Golowich et al., 2018; Neyshabur et al., 2019; Wei and Ma, 2019a; Cao and Gu, 2019; Daniely and Granot, 2019). One interesting point that has been explored, with roots in (Bartlett, 1998), is that even if there are many parameters, the set of models that can be represented using weights with small magnitude is limited enough to provide leverage for induction (Neyshabur et al., 2015; Bartlett et al., 2017; Neyshabur et al., 2018). Intuitively, if the weights start small, since the most popular training algorithms make small, incremental updates that get smaller as the training accuracy improves, there is a tendency for these algorithms to produce small weights. (For some deeper theoretical exploration of implicit bias in deep learning and related settings, see (Gunasekar et al., 2017; 2018a;b; Ma et al., 2018).) Even more recently, authors have proved generalization bounds in terms of the distance from the initial setting of the weights instead of the size of the weights (Dziugaite and Roy, 2017; Bartlett et al., 2017; Neyshabur et al., 2019; Nagarajan and Kolter, 2019). This is important because small initial weights may promote vanishing gradients; it is advisable instead to choose initial weights that maintain a strong but non-exploding signal as computation flows through the network (see LeCun et al., 2012; Glorot and Bengio, 2010; Saxe et al., 2013; He et al., 2015). A number of recent theoretical analyses have shown that, for a large network initialized in this way, accurate models can be found by traveling a short distance in parameter space (see Du et al., 2019b;a; Allen-Zhu et al., 2019; Zou et al., 2018; Lee et al., 2019). Thus, the distance from initialization may be expected to be significantly smaller than the magnitude of the weights. Furthermore, there is theoretical reason to expect that, as the number of parameters increases, the distance from initialization decreases. This motivates generalization bounds in terms of distance from initialization (Dziugaite and Roy, 2017; Bartlett et al., 2017).

Convolutional layers are used in all competitive deep neural network architectures applied to image processing tasks. The most influential generalization analyses in terms of distance from initialization have thus far concentrated on networks with fully connected layers. Since a convolutional layer has an alternative representation as a fully connected layer, these analyses apply in the case of convolutional networks, but, intuitively, the weight-tying employed in the convolutional layer constrains the set of functions computed by the layer. This additional restriction should be expected to aid generalization.

---

[*]Authors are ordered alphabetically.

In this paper, we prove new generalization bounds for convolutional networks that take account of this effect. As in earlier analyses for the fully connected case, our bounds are in terms of the distance from the initial weights, and the number of parameters. Additionally, our bounds independent of the number of pixels in the input, or the height and width of the hidden feature maps.

Our most general bounds apply to networks including both convolutional and fully connected layers, and, as such, they also apply for purely fully connected networks. In contrast with earlier bounds for settings like the one considered here, our bounds are in terms of a sum over layers of the distance from initialization of the layer. Earlier bounds were in terms of product of these distances which led to an exponential dependency on depth. Our bounds have linear dependency on depth which is more aligned with practical observations.

As is often the case for generalization analyses, the central technical lemmas are bounds on covering numbers. Borrowing a technique due to Barron et al. (1999), these are proved by bounding the Lipschitz constant of the mapping from the parameters to the loss of the functions computed by the networks. (Our proof also borrows ideas from the analysis of the fully connected case, especially (Bartlett et al., 2017; Neyshabur et al., 2018).) Covering bounds may be applied to obtain a huge variety of generalization bounds. We present two examples for each covering bound. One is a standard bound on the difference between training and test error. Perhaps the more relevant bound has the flavor of "relative error"; it is especially strong when the training loss is small, as is often the case in modern practice. Our covering bounds are polynomial in the inverse of the granularity of the cover. Such bounds seem to be especially useful for bounding the relative error.

In particular, our covering bounds are of the form $(B/\epsilon)^W$, where $\epsilon$ is the granularity of the cover, $B$ is proportional to the Lipschitz constant of a mapping from parameters to functions, and $W$ is the number of parameters in the model. We apply a bound from the empirical process literature in terms of covering bounds of this form due to Giné and Guillou (2001), who paid particular attention to the dependence of estimation error on $B$. This bound may be helpful for other analyses of the generalization of deep learning in terms of different notions of distance from initialization. (Applying bounds in terms of Dudley's entropy integral in the standard way leads to an exponentially worse dependence on $B$.)

**Related previous work.** Du et al. (2018) proved bounds for CNNs in terms of the number of parameters, for two-layer networks. Arora et al. (2018) analyzed the generalization of networks output by a compression scheme applied to CNNs. Zhou and Feng (2018) provided a generalization guarantee for CNNs satisfying a constraint on the rank of matrices formed from their kernels. Li et al. (2018) analyzed the generalization of CNNs under other constraints on the parameters. Lee and Raginsky (2018) provided a size-free bound for CNNs in a general unsupervised learning framework that includes PCA and codebook learning.

**Related independent work.** Ledent et al. (2019) proved bounds for CNNs that also took account of the effect of weight-tying. (Their bounds retain the exponential dependence on the depth of the network from earlier work, and are otherwise qualitatively dissimilar to ours.) Wei and Ma (2019b) obtained bounds for fully connected networks with an improved dependence on the depth of the network. Daniely and Granot (2019) obtained improved bounds for constant-depth fully-connected networks. Jiang et al. (2019) conducted a wide-ranging empirical study of the dependence of the generalization gap on a variety of quantities, including distance from initialization.

**Notation.** If $K^{(i)}$ is the kernel of convolutional layer number $i$, then $\text{op}(K^{(i)})$ refers to its operator matrix [1] and $\text{vec}(K^{(i)})$ denotes the vectorization of the kernel tensor $K^{(i)}$. For matrix $M$, $\|M\|_2$ denotes the operator norm of $M$. For vectors, $||\cdot||$ represents the Euclidian norm, and $||\cdot||_1$ is the $L_1$ norm. For a multiset $S$ of elements of some set $Z$, and a function $g$ from $Z$ to $\mathbb{R}$, let $\mathbb{E}_S[g] = \frac{1}{m}\sum_{t=1}^{m} g(z_t)$. We will denote the function parameterized by $\Theta$ by $f_\Theta$.

## 2 BOUNDS FOR A BASIC SETTING

In this section, we provide a bound for a clean and simple setting.

---

[1]Convolution is a linear operator and can thus be written as a matrix-vector product. The operator matrix of kernel $K$, refers to the matrix that describes convolving the input with kernel $K$. For details, see (Sedghi et al., 2018).

## 2.1 THE SETTING AND THE BOUNDS

In the basic setting, the input and all hidden layers have the same number $c$ of channels. Each input $x \in \mathbb{R}^{d \times d \times c}$ satisfies $\|\operatorname{vec}(x)\| \leq 1$.

We consider a deep convolutional network, whose convolutional layers use zero-padding (see Goodfellow et al., 2016). Each layer but the last consists of a convolution followed by an activation function that is applied componentwise. The activations are 1-Lipschitz and nonexpansive (examples include ReLU and tanh). The kernels of the convolutional layers are $K^{(i)} \in \mathbb{R}^{k \times k \times c \times c}$ for $i \in \{1, \ldots, L\}$. Let $K = (K^{(1)}, ..., K^{(L)})$ be the $L \times k \times k \times c \times c$-tensor obtained by concatenating the kernels for the various layers. Vector $w$ represents the last layer; the weights in the last layer are fixed with $\|w\| = 1$. Let $W = Lk^2c^2$ be the total number of trainable parameters in the network.

We let $K_0^{(1)}, \ldots, K_0^{(L)}$ take arbitrary fixed values (interpreted as the initial values of the kernels) subject to the constraint that, for all layers $i$, $\|\operatorname{op}(K_0^{(i)})\|_2 = 1$. (This is often the goal of initialization schemes.) Let $K_0$ be the corresponding $L \times k \times k \times c \times c$ tensor. We provide a generalization bound in terms of distance from initialization, along with other natural parameters of the problem. The distance is measured with $\|K - K_0\|_\sigma \stackrel{\text{def}}{=} \sum_{i=1}^L \|\operatorname{op}(K^{(i)}) - \operatorname{op}(K_0^{(i)})\|_2$.

For $\beta > 0$, define $\mathcal{K}_\beta$ to be the set of kernel tensors within $\|\cdot\|_\sigma$ distance $\beta$ of $K_0$, and define $F_\beta$ to be set of functions computed by CNNs with kernels in $\mathcal{K}_\beta$. That is, $F_\beta = \{f_K : \|K - K_0\|_\sigma \leq \beta\}$.

Let $\ell : \mathbb{R} \times \mathbb{R} \to [0, 1]$ be a loss function such that $\ell(\cdot, y)$ is $\lambda$-Lipschitz for all $y$. An example is the $1/\lambda$-margin loss.

For a function $f$ from $\mathbb{R}^{d \times d \times c}$ to $\mathbb{R}$, let $\ell_f(x, y) = \ell(f(x), y)$.

We will use $S$ to denote a set $\{(x_1, y_1), \ldots, (x_m, y_m)\} = \{z_1, \ldots z_m\}$ of random training examples where each $z_t = (x_t, y_t)$.

**Theorem 2.1** (Basic bounds). *For any $\eta > 0$, there is a $C > 0$ such that for any $\beta, \delta > 0$, $\lambda \geq 1$, for any joint probability distribution $P$ over $\mathbb{R}^{d \times d \times c} \times \mathbb{R}$, if a training set $S$ of $n$ examples is drawn independently at random from $P$, then, with probability at least $1 - \delta$, for all $f \in F_\beta$,*

$$\mathbb{E}_{z \sim P}[\ell_f(z)] \leq (1 + \eta)\mathbb{E}_S[\ell_f] + \frac{C(W(\beta + \log(\lambda n)) + \log(1/\delta))}{n}$$

*and, if $\beta \geq 5$, then*

$$\mathbb{E}_{z \sim P}[\ell_f(z)] \leq \mathbb{E}_S[\ell_f] + C\sqrt{\frac{W(\beta + \log(\lambda)) + \log(1/\delta)}{n}}$$

*and otherwise*

$$\mathbb{E}_{z \sim P}[\ell_f(z)] \leq \mathbb{E}_S[\ell_f] + C\left(\beta\lambda\sqrt{\frac{W}{n}} + \sqrt{\frac{\log(1/\delta)}{n}}\right).$$

If Theorem 2.1 is applied with the margin loss, then $\mathbb{E}_{z \sim P}[\ell_f(z)]$ is in turn an upper bound on the probability of misclassification on test data. Using the algorithm from (Sedghi et al., 2018), $\|\cdot\|_\sigma$ may be efficiently computed. Since $\|K - K_0\|_\sigma \leq \|\operatorname{vec}(K) - \operatorname{vec}(K_0)\|_1$ (Sedghi et al., 2018), Theorem 2.1 yields the same bounds as a corollary if the definition of $F_\beta$ is replaced with the analogous definition using $\|\operatorname{vec}(K) - \operatorname{vec}(K_0)\|_1$.

## 2.2 TOOLS

**Definition 2.2.** *For $d \in N$, a set $G$ of real-valued functions with a common domain $Z$, we say that $G$ is $(B, d)$-Lipschitz parameterized if there is a norm $\|\cdot\|$ on $\mathbb{R}^d$ and a mapping $\phi$ from the unit ball w.r.t. $\|\cdot\|$ in $\mathbb{R}^d$ to $G$ such that, for all $\theta$ and $\theta'$ such that $\|\theta\| \leq 1$ and $\|\theta'\| \leq 1$, and all $z \in Z$, $|(\phi(\theta))(z) - (\phi(\theta'))(z)| \leq B\|\theta - \theta'\|$.*

The following lemma is essentially known. Its proof, which uses standard techniques (see Pollard, 1984; Talagrand, 1994; 1996; Barron et al., 1999; Van de Geer, 2000; Giné and Guillou, 2001; Mohri et al., 2018), is in Appendix A.

**Lemma 2.3.** *Suppose a set $G$ of functions from a common domain $Z$ to $[0, M]$ is $(B, d)$-Lipschitz parameterized for $B > 0$ and $d \in \mathbb{N}$.*

*Then, for any $\eta > 0$, there is a $C$ such that, for all large enough $n \in \mathbb{N}$, for any $\delta > 0$, for any probability distribution $P$ over $Z$, if $S$ is obtained by sampling $n$ times independently from $P$, then, with probability at least $1 - \delta$, for all $g \in G$,*

$$\mathbb{E}_{z \sim P}[g(z)] \leq (1 + \eta)\mathbb{E}_S[g] + \frac{CM(d\log(Bn) + \log(1/\delta))}{n}$$

*and if $B \geq 5$,*

$$\mathbb{E}_{z \sim P}[g(z)] \leq \mathbb{E}_S[g] + CM\sqrt{\frac{d\log B + \log\frac{1}{\delta}}{n}},$$

*and, for all $B$,*

$$\mathbb{E}_{z \sim P}[g(z)] \leq \mathbb{E}_S[g] + C\left(B\sqrt{\frac{d}{n}} + M\sqrt{\frac{\log\frac{1}{\delta}}{n}}\right).$$

## 2.3 PROOF OF THEOREM 2.1

**Definition 2.4.** *Let $\ell_F = \{\ell_f : f \in F\}$.*

We will prove Theorem 2.1 by showing that $\ell_{F_\beta}$ is $(\beta\lambda e^\beta, W)$-Lipschitz parameterized. This will be achieved through a series of lemmas.

**Lemma 2.5.** *Choose $K \in \mathcal{K}_\beta$ and a layer $j$. Suppose $\tilde{K} \in \mathcal{K}_\beta$ satisfies $K^{(i)} = \tilde{K}^{(i)}$ for all $i \neq j$. Then, for all examples $(x, y)$, $|\ell(f_K(x), y) - \ell(f_{\tilde{K}}(x), y)| \leq \lambda e^\beta \left\|\operatorname{op}(K^{(j)}) - \operatorname{op}(\tilde{K}^{(j)})\right\|_2$.*

*Proof.* For each layer $i$, let $\beta_i = \|\operatorname{op}(K^{(i)}) - \operatorname{op}(K_0^{(i)})\|_2$.

Since $\ell$ is $\lambda$-Lipschitz w.r.t. its first argument, we have that $|\ell(f_K(x), y) - \ell(f_{\tilde{K}}(x), y)| \leq \lambda|f_K(x) - f_{\tilde{K}}(x)|$, so it suffices to bound $|f_K(x) - f_{\tilde{K}}(x)|$. Let $g_{\text{up}}$ be the function from the inputs to the whole network with parameters $K$ to the inputs to the convolution in layer $j$, and let $g_{\text{down}}$ be the function from the output of this convolution to the output of the whole network, so that $f_K = g_{\text{down}} \circ f_{\operatorname{op}(K^{(j)})} \circ g_{\text{up}}$. Choose an input $x$ to the network, and let $u = g_{\text{up}}(x)$. Recalling that $\|x\| \leq 1$, and the non-linearities are nonexpansive, we have $\|u\| \leq \prod_{i<j} \left\|\operatorname{op}(K^{(i)})\right\|_2$. Since the non-linearities are 1-Lipschitz, and, recalling that $K^{(i)} = \tilde{K}^{(i)}$ for $i \neq j$, we have

$$|f_K(x) - f_{\tilde{K}}(x)| = |g_{\text{down}}(\operatorname{op}(K^{(j)})u) - g_{\text{down}}(\operatorname{op}(\tilde{K}^{(j)})u)|$$

$$\leq \left(\prod_{i>j} \left\|\operatorname{op}(K^{(i)})\right\|_2\right)\left\|\operatorname{op}(K^{(j)})u - \operatorname{op}(\tilde{K}^{(j)})u\right\|$$

$$\leq \left(\prod_{i>j} \left\|\operatorname{op}(K^{(i)})\right\|_2\right)\left\|\operatorname{op}(K^{(j)}) - \operatorname{op}(\tilde{K}^{(j)})\right\|_2 \|u\|$$

$$\leq \left(\prod_{i \neq j} \left\|\operatorname{op}(K^{(i)})\right\|_2\right)\left\|\operatorname{op}(K^{(j)}) - \operatorname{op}(\tilde{K}^{(j)})\right\|_2$$

$$\leq \left(\prod_{i \neq j}(1 + \beta_i)\right)\left\|\operatorname{op}(K^{(j)}) - \operatorname{op}(\tilde{K}^{(j)})\right\|_2$$

where the last inequality uses the fact that $||\operatorname{op}(K^{(i)}) - \operatorname{op}(K_0^{(i)})||_2 \leq \beta_i$ for all $i$ and $||\operatorname{op}(K_0^{(i)})||_2 = 1$ for all $i$.

Now $\prod_{i \neq j}(1 + \beta_i) \leq \prod_{i=1}^{L}(1 + \beta_i)$, and the latter is maximized over the nonnegative $\beta_i$'s subject to $\sum_{i \neq j} \beta_i \leq \beta$ when each of them is $\beta/L$. Since $(1 + \beta/L)^L \leq e^\beta$, this completes the proof. $\square$

Now we prove a bound when all the layers can change between $K$ and $\tilde{K}$.

**Lemma 2.6.** *For any $K, \tilde{K} \in \mathcal{K}_\beta$, for any input $x$ to the network, $|\ell(f_K(x), y) - \ell(f_{\tilde{K}}(x), y)| \leq \lambda e^\beta \left\| K - \tilde{K} \right\|_\sigma$.*

*Proof.* Consider transforming $K$ to $\tilde{K}$ by replacing one layer of $K$ at a time with the corresponding layer in $\tilde{K}$. Applying Lemma 2.5 to bound the distance traversed with each replacement and combining this with the triangle inequality gives

$$|\ell(f_K(x), y) - \ell(f_{\tilde{K}}(x), y)| \leq \lambda e^\beta \sum_{j=1}^{L} \left\| \operatorname{op}(K^{(j)}) - \operatorname{op}(\tilde{K}^{(j)}) \right\|_2 = \lambda e^\beta \left\| K - \tilde{K} \right\|_\sigma.$$

$\square$

Now we are ready to prove our basic bound.

*Proof (of Theorem 2.1).* Consider the mapping $\phi$ from the ball w.r.t. $\left\| \cdot \right\|_\sigma$ of radius 1 in $\mathbb{R}^{Lk^2c^2}$ centered at $\operatorname{vec}(K_0)$ to $\ell_{F_\beta}$ defined by $\phi(\theta) = \ell_{f_{K_0 + \beta \operatorname{vec}^{-1}(\theta)}}$, where $\operatorname{vec}^{-1}(\theta)$ is the reshaping of $\theta$ into a $L \times k \times k \times c \times c$-tensor. Lemma 2.6 implies that this mapping is $\beta \lambda e^\beta$-Lipschitz. Applying Lemma 2.3 completes the proof. $\square$

## 2.4 COMPARISONS

Since a convolutional network has an alternative parameterization as a fully connected network, the bounds of (Bartlett et al., 2017) have consequences for convolutional networks. To compare our bound with this, first, note that Theorem 2.1, together with standard model selection techniques, yields a

$$O\left( \sqrt{\frac{W\left(||K - K_0||_\sigma + \log(\lambda)\right) + \log(1/\delta)}{n}} \right) \quad (1)$$

bound on $\mathbb{E}_{z \sim P}[\ell_f(z)] - \mathbb{E}_S[\ell_f(z)]$ (For more details, please see Appendix B.) Translating the bound of (Bartlett et al., 2017) to our setting and notation directly yields a bound on $\mathbb{E}_{z \sim P}[\ell_f(z)] - \mathbb{E}_S[\ell_f(z)]$ whose main terms are proportional to

$$\frac{\lambda \left( \prod_{i=1}^{L} ||\operatorname{op}(K^{(i)})||_2 \right) \left( \sum_{i=1}^{L} \frac{||\operatorname{op}(K^{(i)})^\top - \operatorname{op}(K_0^{(i)})^\top||_{2,1}^{2/3}}{||\operatorname{op}(K^{(i)})||_2^{2/3}} \right)^{3/2} \log(d^4 c^2 L) + \sqrt{\log(1/\delta)}}{\sqrt{n}} \quad (2)$$

where, for a $p \times q$ matrix $A$, $||A||_{2,1} = ||(||A_{:,1}||_2, ..., ||A_{:,q}||_2)||_1$. One can get an idea of how this bound relates to (1) by comparing the bounds in a simple concrete case. Suppose that each of the convolutional layers of the network parameterized by $K_0$ computes the identity function, and that $K$ is obtained from $K_0$ by adding $\epsilon$ to each entry. In this case, disregarding edge effects, for all $i$, $||\operatorname{op}(K^{(i)})||_2 = 1 + \epsilon k^2 c$ and $||K - K_0||_\sigma = \epsilon k^2 cL$ (as proved in Appendix C). Also, $||\operatorname{op}(K^{(i)})^\top - \operatorname{op}(K_0^{(i)})^\top||_{2,1} = (cd^2)(\epsilon\sqrt{ck^2}) = \epsilon c^{3/2} d^2 k$. We get additional simplification if we set $\epsilon = \frac{1}{k^2}$. In this case, (2) gives a constant times

$$\frac{(c + 1)^L \sqrt{c} d(d/k) L^{3/2} \lambda \log(dcL) + \sqrt{\log(1/\delta)}}{\sqrt{n}}$$

where (1) gives a constant times

$$\frac{c^{3/2}kL + ck\sqrt{\log(\lambda)} + \sqrt{\log(1/\delta)}}{\sqrt{n}}.$$

In this scenario, the new bound is independent of $d$, and grows more slowly with $\lambda$, $c$ and $L$. Note that $k \le d$ (and, typically, it is much less).

This specific case illustrates a more general phenomenon that holds when the initialization is close to the identity, and changes to the parameters are on a similar scale.

Golowich et al. (2017) established bounds that improve on the bounds of (Bartlett et al., 2017) in some cases. Their bound requires a restriction on the activation function (albeit one that is satisfied by the ReLU). For large $n$, the main term of the natural consequence of Corollary 1 of their paper in the setting of this section, with the required additional assumption on the activation function, grows like

$$\frac{\lambda\sqrt{L}\prod_{i=1}^{L}||\operatorname{op}(K^{(i)})||_F}{\sqrt{n}} \approx \frac{\lambda(cd)^L\sqrt{L}}{\sqrt{n}},$$

when $\epsilon = 1/k^2$. (We note, however, that, in addition to not trying to take account of the convolutional structure, Golowich et al. (2017) also did not make an effort to obtain stronger bounds in the case that the distance from initialization is small. On the other hand, we suspect that modifying their proof analogously to (Bartlett et al., 2017) to do so would not remove the exponential dependence on $L$.)

## 3 A MORE GENERAL BOUND

In this section, we generalize Theorem 2.1.

### 3.1 THE SETTING

The more general setting concerns a neural network where the input is a $d \times d \times c$ tensor whose flattening has Euclidian norm at most $\chi$, and network's output is a $m$-dimensional vector, which may be logits for predicting a one-hot encoding of an $m$-class classification problem.

The network is comprised of $L_c$ convolutional layers followed by $L_f$ fully connected layers. The $i$th convolutional layer includes a convolution, with kernel $K^{(i)} \in \mathbb{R}^{k_i \times k_i \times c_{i-1} \times c_i}$, followed by a componentwise non-linearity and an optional pooling operation. We assume that the non-linearity and any pooling operations are 1-Lipschitz and nonexpansive. Let $V^{(i)}$ be the matrix of weights for the $i$th fully connected layer. Let $\Theta = (K^{(1)}, ..., K^{(L_c)}, V^{(1)}, ..., V^{(L_f)})$ be all of the parameters of the network. Let $L = L_c + L_f$.

We assume that, for all $y$, $\ell(\cdot, y)$ is $\lambda$-Lipschitz for all $y$ and that $\ell(\hat{y}, y) \in [0, M]$ for all $\hat{y}$ and $y$.

An example $(x, y)$ includes a $d \times d \times c$-tensor $x$ and $y \in \mathbb{R}^m$.

We let $K_0^{(1)}, \ldots, K_0^{(L_c)}, V_0^{(1)}, ..., V_0^{(L_f)}$ take arbitrary fixed values subject to the constraint that, for all convolutional layers $i$, $||\operatorname{op}(K_0^{(i)})||_2 \le 1+\nu$, and for all fully connected layers $i$, $||V_0^{(i)}||_2 \le 1+\nu$. Let $\Theta_0 = (K_0^{(1)}, ..., K_0^{(L_c)}, V_0^{(1)}, ..., V_0^{(L_f)})$.

For $\Theta = (K^{(1)}, ..., K^{(L_c)}, V^{(1)}, ..., V^{(L_f)})$ and $\tilde{\Theta} = (\tilde{K}^{(1)}, ..., \tilde{K}^{(L_c)}, \tilde{V}^{(1)}, ..., \tilde{V}^{(L_f)})$. define

$$||\Theta - \tilde{\Theta}||_N = \left(\sum_{i=1}^{L_c} ||\operatorname{op}(K^{(i)}) - \operatorname{op}(\tilde{K}^{(i)})||_2\right) + \sum_{i=1}^{L_f} ||V^{(i)} - \tilde{V}^{(i)}||_2.$$

For $\beta, \nu \ge 0$, define $\mathcal{F}_{\beta,\nu}$ to be set of functions computed by CNNs as described in this subsection with parameters within $|| \cdot ||_N$-distance $\beta$ of $\Theta_0$. Let $\mathcal{O}_{\beta,\nu}$ be the set of their parameterizations.

**Theorem 3.1** (General Bound). *For any $\eta > 0$, there is a constant $C$ such that the following holds. For any $\beta, \nu, \chi > 0$, for any $\delta > 0$, for any joint probability distribution $P$ over $\mathbb{R}^{d \times d \times c} \times \mathbb{R}^m$ such that, with probability 1, $(x, y) \sim P$ satisfies $||\operatorname{vec}(x)||_2 \le \chi$, under the assumptions of this section,*

*if a training set $S$ of $n$ examples is drawn independently at random from $P$, then, with probability at least $1 - \delta$, for all $f \in \mathcal{F}_{\beta,\nu}$,*

$$\mathbb{E}_{z \sim P}[\ell_f(z)] \leq (1 + \eta)\mathbb{E}_S[\ell_f] + \frac{CM\left(W\left(\beta + \nu L + \log(\chi\lambda\beta n)\right) + \log(1/\delta)\right)}{n}$$

*and, if $\chi\lambda\beta(1 + \nu + \beta/L)^L \geq 5$,*

$$\mathbb{E}_{z \sim P}[\ell_f(z)] \leq \mathbb{E}_S[\ell_f] + CM\sqrt{\frac{W\left(\beta + \nu L + \log(\chi\lambda\beta)\right) + \log(1/\delta)}{n}}$$

*and a bound of*

$$\mathbb{E}_{z \sim P}[\ell_f(z)] \leq \mathbb{E}_S[g] + C\left(\chi\lambda\beta(1 + \nu + \beta/L)^L\sqrt{\frac{W}{n}} + M\sqrt{\frac{\log\frac{1}{\delta}}{n}}\right)$$

*holds for all $\chi, \lambda, \beta > 0$.*

## 3.2 Proof of Theorem 3.1

We will prove Theorem 3.1 by using $||\cdot||_N$ to witness the fact that $\ell_{\mathcal{F}_{\beta,\nu}}$ is $\left(\chi\lambda\beta(1 + \nu + \beta/L)^L, W\right)$-Lipschitz parameterized.

The first two lemmas concern the effect of changing a single layer. Their proofs are very similar to the proof of Lemma 2.5, and are in the Appendices D and E.

**Lemma 3.2.** *Choose* $\Theta = (K^{(1)}, ..., K^{(L_c)}, V^{(1)}, ..., V^{(L_f)}), \tilde{\Theta} = (\tilde{K}^{(1)}, ..., \tilde{K}^{(L_c)}, \tilde{V}^{(1)}, ..., \tilde{V}^{(L_f)}) \in \mathcal{O}_{\beta,\nu}$ *and a convolutional layer $j$. Suppose that $K^{(i)} = \tilde{K}^{(i)}$ for all convolutional layers $i \neq j$ and $V^{(i)} = \tilde{V}^{(i)}$ for all fully connected layers $i$. Then, for all examples $(x, y)$,*

$$|\ell(f_\Theta(x), y) - \ell(f_{\tilde{\Theta}}(x), y)| \leq \chi\lambda(1 + \nu + \beta/L)^L \left\|\mathrm{op}(K^{(j)}) - \mathrm{op}(\tilde{K}^{(j)})\right\|_2.$$

**Lemma 3.3.** *Choose* $\Theta = (K^{(1)}, ..., K^{(L_c)}, V^{(1)}, ..., V^{(L_f)}), \tilde{\Theta} = (\tilde{K}^{(1)}, ..., \tilde{K}^{(L_c)}, \tilde{V}^{(1)}, ..., \tilde{V}^{(L_f)}) \in \mathcal{O}_{\beta,\nu}$ *and a fully connected layer $j$. Suppose that $K^{(i)} = \tilde{K}^{(i)}$ for all convolutional layers $i$ and $V^{(i)} = \tilde{V}^{(i)}$ for all fully connected layers $i \neq j$. Then, for all examples $(x, y)$,*

$$|\ell(f_\Theta(x), y) - \ell(f_{\tilde{\Theta}}(x), y)| \leq \chi\lambda(1 + \nu + \beta/L)^L \left\|V^{(j)} - \tilde{V}^{(j)}\right\|_2.$$

Now we prove a bound when all the layers can change between $\Theta$ and $\tilde{\Theta}$.

**Lemma 3.4.** *For any* $\Theta = (K^{(1)}, ..., K^{(L_c)}, V^{(1)}, ..., V^{(L_f)}), \tilde{\Theta} = (\tilde{K}^{(1)}, ..., \tilde{K}^{(L_c)}, \tilde{V}^{(1)}, ..., \tilde{V}^{(L_f)}) \in \mathcal{O}_{\beta,\nu}$, *for any input $x$,*

$$|\ell(f_\Theta(x), y) - \ell(f_{\tilde{\Theta}}(x), y)| \leq \chi\lambda(1 + \nu + \beta/L)^L \left\|\Theta - \tilde{\Theta}\right\|_N.$$

*Proof.* Consider transforming $\Theta$ to $\tilde{\Theta}$ by replacing one layer at a time of $\Theta$ with the corresponding layer in $\tilde{\Theta}$. Applying Lemma 3.2 to bound the distance traversed with each replacement of a convolutional layer, and Lemma 3.3 to bound the distance traversed with each replacement of a fully connected layer, and combining this with the triangle inequality gives the lemma. $\square$

Now we are ready to prove our more general bound.

*Proof (of Theorem 3.1).* Consider the mapping $\phi$ from the ball of $||\cdot||_N$-radius 1 centered at $\Theta_0$ to $\ell_{\mathcal{F}_{\beta,\nu}}$ defined by $\phi(\Theta) = \ell_{f_{\Theta_0 + \beta\Theta}}$. Lemma 2.6 implies that this mapping is $\left(\chi\lambda\beta(1 + \nu + \beta/L)^L, W\right)$-Lipschitz. Applying Lemma 2.3 completes the proof. $\square$

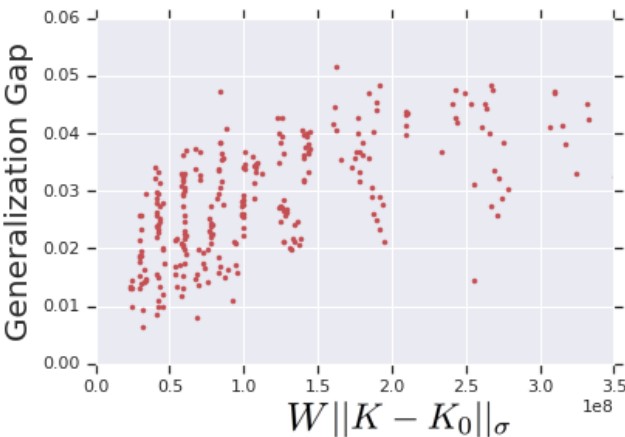

Figure 1: Generalization gaps for a 10-layer all-conv model on CIFAR10 dataset.

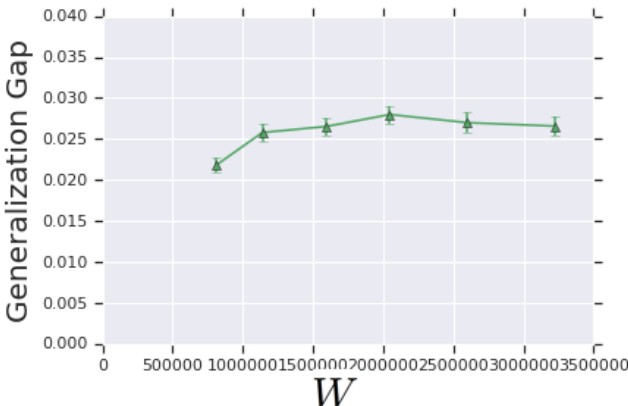

Figure 2: Generalization gap as a function of $W$

### 3.3 MORE COMPARISONS

Theorem 3.1 applies in the case that there are no convolutional layers, i.e. for a fully connected network. In this subsection, we compare its bound in this case with the bound of (Bartlett et al., 2017). Because the bounds are in terms of different quantities, we compare them in a simple concrete case. In this case, for $D = cd^2$, each hidden layer has $D$ components, and there are $D$ classes. For all $i$, $V_0^{(i)} = I$ and $V^{(i)} = I + H/\sqrt{D}$, where $H$ is a Hadamard matrix (using the Sylvester construction), and $\chi = M = 1$. Then, dropping the superscripts, each layer $V$ has $||V||_2 = 2$, $||V - V_0||_2 = 1$, $||V - V_0||_{2,1} = D$.

Further, in the notation of Theorem 3.1, $W = D^2 L$, and $\beta = L$, and $\nu = 0$. Plugging into to Theorem 3.1 yields a bound on the generalization gap proportional to

$$\frac{DL + D\sqrt{L\log(\lambda)} + \sqrt{\log(1/\delta)}}{\sqrt{n}}$$

where, in this case, the bound of (Bartlett et al., 2017) is proportional to

$$\frac{\lambda 2^L L^{3/2} D\log(DL) + \sqrt{\log(1/\delta)}}{\sqrt{n}}$$

and, when $D$ is large relative to $L$, Corollary 1 of (Golowich et al., 2017) (approximately) gives

$$\frac{\lambda 2^{L/2}\sqrt{L}D^{L/2} + \sqrt{\log(1/\delta)}}{\sqrt{n}}.$$

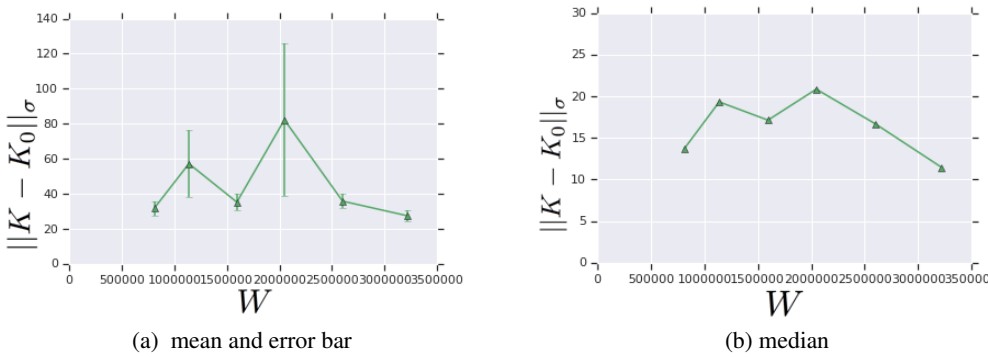

(a) mean and error bar  (b) median

Figure 3: $||K - K_0||_\sigma$ as a function of $W$.

## 4 EXPERIMENTS

We trained a 10-layer all-convolutional model on the CIFAR-10 dataset. The architecture was similar to VGG (Simonyan and Zisserman, 2014). The network was trained with dropout regularization and an exponential learning rate schedule. We define the generalization gap as the difference between train error and test error. In order to analyze the effect of the number of network parameters on the generalization gap, we scaled up the number of channels in each layer, while keeping other elements of the architecture, including the depth, fixed. Each network was trained repeatedly, sweeping over different values of the initial learning rate and batch sizes $32, 64, 128$. For each setting the results were averaged over five different random initializations. Figure 1 shows the generalization gap for different values of $W||K - K_0||_\sigma$. As in the bound of Theorem 3.1, the generalization gap increases with $W||K - K_0||_\sigma$. Figure 2 shows that as the network becomes more over-parametrized, the generalization gap remains almost flat with increasing $W$. This is expected due to role of over-parametrization on generalization (Neyshabur et al., 2019). An explanation of this phenomenon that is consistent with the bound presented here is that, ultimately, increasing $W$ leads to a decrease in value of $||K - K_0||_\sigma$; see Figure 3a. The fluctuations in Figure 3a are partly due to the fact that training neural networks is not an stable process. We provide the medians $||K - K_0||_\sigma$ for different values of $W$ in Figure 3b.

### ACKNOWLEDGMENTS

We thank Peter Bartlett, Jaeho Lee and Sam Schoenholz for valuable conversations, and anonymous reviewers for helpful comments.

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

# A    PROOF OF LEMMA 2.3

**Definition A.1.** *If $(X, \rho)$ is a metric space and $H \subseteq X$, we say that $G$ is an $\epsilon$-cover of $H$ with respect to $\rho$ if every $h \in H$ has a $g \in G$ such that $\rho(g, h) \leq \epsilon$. Then $\mathcal{N}_\rho(H, \epsilon)$ denotes the size of the smallest $\epsilon$-cover of $H$ w.r.t. $\rho$.*

**Definition A.2.** *For a domain $Z$, define a metric $\rho_{\max}$ on pairs of functions from $Z$ to $\mathbb{R}$ by $\rho_{\max}(f, g) = \sup_{z \in Z} |f(z) - g(z)|$.*

We need two lemmas in terms of these covering numbers. The first is by now a standard bound from Vapnik-Chervonenkis theory (Vapnik and Chervonenkis, 1971; Vapnik, 1982; Pollard, 1984). For example, it is a direct consequence of (Haussler, 1992, Theorem 3).

**Lemma A.3.** *For any $\eta > 0$, there is a constant $C$ depending only on $\eta$ such that the following holds. Let $G$ be an arbitrary set of functions from a common domain $Z$ to $[0, M]$. If there are constants $B$ and $d$ such that, $\mathcal{N}_{\rho_{\max}}(G, \epsilon) \leq \left(\frac{B}{\epsilon}\right)^d$ for all $\epsilon > 0$, then, for all large enough $n \in \mathbb{N}$, for any $\delta > 0$, for any probability distribution $P$ over $Z$, if $S$ is obtained by sampling $n$ times independently from $P$, then, with probability at least $1 - \delta$, for all $g \in G$,*

$$\mathbb{E}_{z \sim P}[g(z)] \leq (1 + \eta)\mathbb{E}_S[g] + \frac{CM(d\log(Bn) + \log(1/\delta))}{n}.$$

We will also use the following, which is the combination of (2.5) and (2.7) of (Giné and Guillou, 2001).

**Lemma A.4.** *Let $G$ be an arbitrary set of functions from a common domain $Z$ to $[0, M]$. If there are constants $B \geq 5$ and $d$ such that $\mathcal{N}_{\rho_{\max}}(G, \epsilon) \leq \left(\frac{B}{\epsilon}\right)^d$ for all $\epsilon > 0$, for all large enough $n \in \mathbb{N}$, for any $\delta > 0$, for any probability distribution $P$ over $Z$, if $S$ is obtained by sampling $n$ times independently from $P$, then, with probability at least $1 - \delta$, for all $g \in G$,*

$$\mathbb{E}_{z \sim P}[g(z)] \leq \mathbb{E}_S[g] + CM\sqrt{\frac{d\log B + \log\frac{1}{\delta}}{n}},$$

*where $C$ is an absolute constant.*

The above bound only holds for $B \geq 5$. The following, which can be obtained by combining Talagrand's Lemma with the standard bound on Rademacher complexity in terms of the Dudley entropy integral (see (Van de Geer, 2000; Bartlett, 2013)), yields a bound for all $B$.

**Lemma A.5.** *Let $G$ be an arbitrary set of functions from a common domain $Z$ to $[0, M]$. If there are constants $B > 0$ and $d$ such that $\mathcal{N}_{\rho_{\max}}(G, \epsilon) \leq \left(\frac{B}{\epsilon}\right)^d$ for all $\epsilon > 0$, then, for all large enough $n \in \mathbb{N}$, for any $\delta > 0$, for any probability distribution $P$ over $Z$, if $S$ is obtained by sampling $n$ times independently from $P$, then, with probability at least $1 - \delta$, for all $g \in G$,*

$$\mathbb{E}_{z \sim P}[g(z)] \leq \mathbb{E}_S[g] + C\left(B\sqrt{\frac{d}{n}} + M\sqrt{\frac{\log\frac{1}{\delta}}{n}}\right),$$

*where $C$ is an absolute constant.*

So now we want a bound on $\mathcal{N}_{\rho_{\max}}(G, \epsilon)$ for Lipschitz-parameterized classes. For this, we need the notion of a packing which we now define.

**Definition A.6.** *For any metric space $(X, \rho)$ and any $H \subseteq S$, let $\mathcal{M}_\rho(H, \epsilon)$ be the size of the largest subset of $H$ whose members are pairwise at a distance greater than $\epsilon$ w.r.t. $\rho$.*

**Lemma A.7** ((Kolmogorov and Tikhomirov, 1959)). *For any metric space $(X, \rho)$, any $H \subseteq X$, and any $\epsilon > 0$, we have*

$$\mathcal{N}_\rho(H, \epsilon) \leq \mathcal{M}_\rho(H, \epsilon).$$

We will also need a lemma about covering a ball by smaller balls. This is probably also already known, and uses a standard proof (see Pollard, 1990, Lemma 4.1), but we haven't found a reference for it.

**Lemma A.8.** *Let*

- *$d$ be a positive integer,*

- *$|| \cdot ||$ be a norm*

- *$\rho$ be the metric induced by $|| \cdot ||$, and*

- *$\kappa, \epsilon > 0$.*

*A ball in $\mathbb{R}^d$ of radius $\kappa$ w.r.t. $\rho$ can be covered by $\left(\frac{3\kappa}{\epsilon}\right)^d$ balls of radius $\epsilon$.*

*Proof.* We may assume without loss of generality that $\kappa > \epsilon$. Let $q > 0$ be the volume of the unit ball w.r.t. $\rho$ in $\mathbb{R}^d$. Then the volume of any $\alpha$-ball with respect to $\rho$ is $\alpha^d q$. Let $B$ be the ball of radius $r$ in $\mathbb{R}^d$. The $\epsilon/2$-balls centered at the members of any $\epsilon$-packing of $B$ are disjoint. Since these centers are contained in $B$, the balls are contained in a ball of radius $\kappa + \epsilon/2$. Thus

$$\mathcal{M}_\rho(B, \epsilon) \left(\frac{\epsilon}{2}\right)^d q \leq \left(\kappa + \frac{\epsilon}{2}\right)^d q \leq \left(\frac{3\kappa}{2}\right)^d q.$$

Solving for $\mathcal{M}_\rho(B, \epsilon)$ and applying Lemma A.7 completes the proof. $\square$

We now prove Lemma 2.3. Let $|| \cdot ||$ be the norm witnessing the fact that $G$ is $(B, d)$-Lipschitz parameterized, and let $\mathcal{B}$ be the unit ball in $\mathbb{R}^d$ w.r.t. $|| \cdot ||$ and let $\rho$ be the metric induced by $|| \cdot ||$. Then, for any $\epsilon$, an $\epsilon/B$-cover of $\mathcal{B}$ w.r.t. $\rho$ induces an $\epsilon$-cover of $G$ w.r.t. $\rho_{\max}$, so

$$N_{\rho_{\max}}(G, \epsilon) \leq N_\rho(\mathcal{B}, \epsilon/B).$$

Applying Lemma A.8, this implies

$$N_{\rho_{\max}}(G, \epsilon) \leq \left(\frac{3B}{\epsilon}\right)^d.$$

Then applying Lemma A.3, Lemma A.4 and Lemma A.5 completes the proof.

## B  PROOF OF (1)

For $\delta > 0$, and for each $j \in \mathbb{N}$, let $\beta_j = 5 \times 2^j$ let $\delta_j = \frac{1}{2j^2}$. Taking a union bound over an application of Theorem 2.1 for each value of $j$, with probability at least $1 - \sum_j \delta_j \geq 1 - \delta$, for all $j$, and all $f \in F_{\beta_j}$

$$\mathbb{E}_{z \sim P}[\ell_f(z)] \leq (1 + \eta)\mathbb{E}_S[\ell_f(z)] + \frac{C(W(\beta_j + \log(\lambda n)) + \log(j/\delta))}{n}$$

and

$$\mathbb{E}_{z \sim P}[\ell_f(z)] \leq \mathbb{E}_S[\ell_f(z)] + C\sqrt{\frac{W(\beta_j + \log(\lambda)) + \log(j/\delta)}{n}}.$$

For any $K$, if we apply these bounds in the case of the least $j$ such that $||K - K_0||_\sigma \leq \beta_j$, we get

$$\mathbb{E}_{z \sim P}[\ell_f(z)] \leq (1 + \eta)\mathbb{E}_S[\ell_f(z)] + \frac{C(W(2||K - K_0||_\sigma + \log(\lambda n)) + \log(\log(||K - K_0||_\sigma)/\delta))}{n}$$

and

$$\mathbb{E}_{z \sim P}[\ell_f(z)] \leq \mathbb{E}_S[\ell_f(z)] + C\sqrt{\frac{W(2||K - K_0||_\sigma + \log(\lambda)) + \log(\log(||K - K_0||_\sigma)/\delta)}{n}},$$

and simplifying completes the proof.

## C  THE OPERATOR NORM OF $\mathrm{op}(K^{(i)})$

Let $J = K^{(i)} - K_0^{(i)}$. Since $||\mathrm{op}(K^{(i)})||_2 = 1 + ||\mathrm{op}(J)||_2$, it suffices to find $||\mathrm{op}(J)||_2$.

For the rest of this section, we number indices from 0, let $[d] = \{0, ..., d-1\}$, and define $\omega = \exp(2\pi i/d)$. To facilitate the application of matrix notation, pad the $k \times k \times c \times c$ tensor $J$ out with zeros to make a $d \times d \times c \times c$ tensor $\tilde{J}$.

The following lemma is an immediate consequence of Theorem 6 of Sedghi et al. (2018).

**Lemma C.1** (Sedghi et al. (2018)). *Let $F$ be the complex $d \times d$ matrix defined by $F_{ij} = \omega^{ij}$.*

*For each $u, v \in [d] \times [d]$, let $P^{(u,v)}$ be the $c \times c$ matrix given by $P_{k\ell}^{(u,v)} = (F^T \tilde{J}_{:,:,k,\ell} F)_{uv}$. Then*

$$||\mathrm{op}(J)||_2 = \max_{u,v} ||P^{(u,v)}||_2.$$

First, note that, by symmetry, for each $u$ and $v$, all components of $P^{(u,v)}$ are the same. Thus,

$$||P^{(u,v)}||_2 = c|P_{00}^{(u,v)}|. \tag{3}$$

For any $u, v$,

$$|P_{00}^{(u,v)}| = \left|\sum_{p,q} \omega^{up} \omega^{vq} \tilde{J}_{p,q,0,0}\right| \leq \epsilon k^2$$

and $P_{00}^{(0,0)} = \epsilon k^2$. Combining this with (3) and Lemma C.1, $||\mathrm{op}(J)||_2 = \epsilon c k^2$, which implies $||\mathrm{op}(K)||_2 = 1 + \epsilon c k^2$.

## D  PROOF OF LEMMA 3.2

For each convolutional layer $i$, let $\beta_i = ||\mathrm{op}(K^{(i)}) - \mathrm{op}(K_0^{(i)})||_2$, and, for each fully connected layer $i$, let $\gamma_i = ||V^{(i)} - V_0^{(i)}||_2$.

Since $\ell$ is $\lambda$-Lipschitz w.r.t. its first argument, we have that $|\ell(f_\Theta(x), y) - \ell(f_{\tilde{\Theta}}(x), y)| \leq \lambda|f_\Theta(x) - f_{\tilde{\Theta}}(x)|$. Let $g_{\mathrm{up}}$ be the function from the inputs to the whole network with parameters $\Theta$ to the inputs to the convolution in layer $j$, and let $g_{\mathrm{down}}$ be the function from the output of this convolution to the output of the whole network, so that $f_\Theta = g_{\mathrm{down}} \circ f_{\mathrm{op}(K^{(j)})} \circ g_{\mathrm{up}}$. Choose an input $x$ to the network, and let $u = g_{\mathrm{up}}(x)$. Recalling that $||x|| \leq \chi$, and that the non-linearities and pooling operations are non-expansive, we have $||u|| \leq \chi \prod_{i<j} \left|\left|\mathrm{op}(K^{(i)})\right|\right|_2$. Using the fact that the non-linearities are

1-Lipschitz, we have

$$|f_\Theta(x) - f_{\tilde\Theta}(x)| = |g_{\text{down}}(\text{op}(K^{(j)})u) - g_{\text{down}}(\text{op}(\tilde K^{(j)})u)|$$

$$\leq \left(\prod_{i>j}\left\|\text{op}(K^{(i)})\right\|_2\right)\left(\prod_i\left\|V^{(i)}\right\|_2\right)\left\|\text{op}(K^{(j)})u - \text{op}(\tilde K^{(j)})u\right\|$$

$$\leq \left(\prod_{i>j}\left\|\text{op}(K^{(i)})\right\|_2\right)\left(\prod_i\left\|V^{(i)}\right\|_2\right)\left\|\text{op}(K^{(j)}) - \text{op}(\tilde K^{(j)})\right\|_2\|u\|$$

$$\leq \chi\left(\prod_{i\neq j}\left\|\text{op}(K^{(i)})\right\|_2\right)\left(\prod_i\left\|V^{(i)}\right\|_2\right)\left\|\text{op}(K^{(j)}) - \text{op}(\tilde K^{(j)})\right\|_2$$

$$\leq \chi\left(\prod_{i\neq j}(1+\nu+\beta_i)\right)\left(\prod_i(1+\nu+\gamma_i)\right)\left\|\text{op}(K^{(j)}) - \text{op}(\tilde K^{(j)})\right\|_2$$

where the last inequality uses the fact that $||\text{op}(K^{(i)}) - \text{op}(K_0^{(i)})||_2 \leq \beta_i$ for all $i$, $||V^{(i)} - V_0^{(i)}||_2 \leq \gamma_i$ for all $i$, $||\text{op}(K_0^{(i)})||_2 \leq 1+\nu$ for all $i$ and $||V_0^{(i)}||_2 \leq 1+\nu$ for all $i$.

Since $\left(\prod_{i\neq j}(1+\nu+\beta_i)\right)\left(\prod_i(1+\nu+\gamma_i)\right) \leq \left(\prod_i(1+\nu+\beta_i)\right)\left(\prod_i(1+\nu+\gamma_i)\right)$, and the latter is maximized subject to $(\sum_i \beta_i) + \sum_i \gamma_i \leq \beta$ when each summand is $\beta/L$, this completes the proof.

## E    PROOF OF LEMMA 3.3

For each convolutional layer $i$, let $\beta_i = ||\text{op}(K^{(i)}) - \text{op}(K_0^{(i)})||_2$, and, for each fully connected layer $i$, let $\gamma_i = ||V^{(i)} - V_0^{(i)}||_2$.

Since $\ell$ is $\lambda$-Lipschitz w.r.t. its first argument, we have that $|\ell(f_\Theta(x), y) - \ell(f_{\tilde\Theta}(x), y)| \leq \lambda|f_\Theta(x) - f_{\tilde\Theta}(x)|$. Let $g_{\text{up}}$ be the function from the inputs to the whole network with parameters $\Theta$ to the inputs to fully connected layer layer $j$, and let $g_{\text{down}}$ be the function from the output of this layer to the output of the whole network, so that $f_\Theta = g_{\text{down}} \circ f_{V^{(j)}} \circ g_{\text{up}}$. Choose an input $x$ to the network, and let $u = g_{\text{up}}(x)$. Recalling that $||x|| \leq \chi$, and that the non-linearities and pooling operations are non-expansive, we have $\|u\| \leq \chi\left(\prod_i\left\|\text{op}(K^{(i)})\right\|_2\right)\left(\prod_{i<j}\left\|V^{(i)}\right\|_2\right)$. Thus

$$|f_\Theta(x) - f_{\tilde\Theta}(x)| = |g_{\text{down}}(V^{(j)}u) - g_{\text{down}}(\tilde V^{(j)}u)|$$

$$\leq \left(\prod_{i>j}\left\|V^{(i)}\right\|_2\right)\left\|V^{(j)}u - \tilde V^{(j)}u\right\|$$

$$\leq \left(\prod_{i>j}\left\|V^{(i)}\right\|_2\right)\left\|V^{(j)} - \tilde V^{(j)}\right\|_2\|u\|$$

$$\leq \chi\left(\prod_i\left\|\text{op}(K^{(i)})\right\|_2\right)\left(\prod_{i\neq j}\left\|V^{(i)}\right\|_2\right)\left\|V^{(j)} - \tilde V^{(j)}\right\|_2$$

$$\leq \chi\left(\prod_i(1+\nu+\beta_i)\right)\left(\prod_{i\neq j}(1+\nu+\gamma_i)\right)\left\|V^{(j)} - \tilde V^{(j)}\right\|_2.$$

Since $\left(\prod_i(1+\nu+\beta_i)\right)\left(\prod_{i\neq j}(1+\nu+\gamma_i)\right) \leq \left(\prod_i(1+\nu+\beta_i)\right)\left(\prod_i(1+\nu+\gamma_i)\right)$, and the latter is maximized subject to $(\sum_i \beta_i) + \sum_i \gamma_i \leq \beta$ when each summand is $\beta/L$, this completes the proof.

