# OpenReview forum: "Generalization bounds for deep convolutional neural networks"
_ICLR.cc/2020/Conference — Accept (Poster)_

### Official Review · AnonReviewer2 · 2019-10-23
**Official Blind Review #2**

**Rating:** 3

**Review:**

Summary

This paper studied the generalization power of CNNs and showed several upper bounds of generalization errors. Their results have two characteristics. First, the bounds are in terms of the quantity that is independent of the input dimension (size-free). Second, the upper bounds involve the distance between initial and learned parameters. These results improved the upper bounds that we can derive by naively applying the results of Bartlett et al. (2017) or Neushubar et al. (2017), because the dominant term of the existing upper bounds contained $l_{2, 1}$ or $l_2$ norms, which could depend on the input dimensions in the worst case. The authors empirically showed that there is a correlation between the generalization error of learned CNNs and the dominant term of the upper bound (i.e., the product of the parameter size and the distance from the set of initial parameters).


Decision

To the best of my knowledge, this is the first work that proved the size-free generalization bound for multi-layer CNNs. However, I think the assumption on the hypothesis class is very restrictive and significantly eases the problem, as I discuss in detail later. Therefore, I judge the technical contribution of the paper is moderate and recommend to reject the paper weakly.
By the standard argument of the statistical learning theory (such as Theorem A.4), we can typically bound the generalization error by $O(B\sqrt{D/N})$ where $B$ is the infimum of Lipschitz constant of hypotheses, $D$ is the intrinsic dimension of the hypothesis class, and $N$ is the sample size. Therefore, we can derive the size-free generalization bound if $B$ does not depend on the input dimension. Since the hypothesis class $F_\beta$ is defined via the spectral norm of CNNs, it is not surprising that we can derive the size-freeness of $B$. The size-free generalization bound has been already proven by Du et al., (2017), although it was the two-layered case. They imposed a restricted eigenvalue assumption. I think it implies that we need more sophisticated analysis if we do not assume the size-freeness of the hypothesis class.


Comments

- The authors claimed that Figure 3 is consistent with theorems because, according to the upper bound of theorems, the distance from the initialization point decreases when the generalization error is the same and the parameter size increases. However, I think it is too aggressive to conclude it from Figure 3 because the decreasing trend in the value of $\|K-K_0\|_\sigma$ is found only around $2\times 10^6\leq W \leq 3\times 10^6$. Furthermore, the value of $\|K-K_0\|_\sigma$ for  $W\approx 5\times 10^5$ is approximately the same as the value for $W\approx 3\times 10^6$.


Suggestions

- Please add the conclusion section which summarizes the paper and discusses the possible research directions.


Minor Comments

- page 1, section 1, paragraph 1
	- ... with roots in (Bartett, 1998) , is that ... → Use \citet
- page 2, section 2.1, paragraph 2
	- Write the definition of "expansive" activations.
- page 3, section 2., theorem 2.1
	- I think we should replace $\log(\lambda n)$ and $\log(\lambda)$ in equations with $\log(\beta \lambda n)$ and $\log(\beta \lambda)$, respectively.
- page 3, section 2.2, definition 2.2
	- $N$ → $\mathbb{N}$

**Experience Assessment:**

I have published one or two papers in this area.

**Review Assessment: Checking Correctness Of Derivations And Theory:**

I carefully checked the derivations and theory.

**Review Assessment: Checking Correctness Of Experiments:**

I assessed the sensibility of the experiments.

**Review Assessment: Thoroughness In Paper Reading:**

I read the paper at least twice and used my best judgement in assessing the paper.

---

> ### Author Response · Authors · 2019-11-07
> **response to AnonReviewer2**
>
> Thank you for your review.
>
> As we wrote in our response to Reviewer 1, if one just applies a B sqrt(D/N) bound, even if the operator norms of all of the layers are bounded by a constant, the dependence on the bound is exponential in the depth of the network.  One contribution of this paper is to recognize that the bounds of Gine and Guillou are especially relevant to deep networks, enabling a linear dependence on the depth.  As far as we know, we are the first authors to do this.
>
> Second, we would like to argue that if the bounds of this paper were obvious, the authors of the NeurIPS’18 paper treating two-layer convolutional networks would have added them to their paper.
>
> Your point regarding the discussion of Figure 3 is well-taken — the distance from initialization only starts trending down once the networks get pretty big.  We will update our paper to describe this finding in a more refined way.
>
> An activation function that is applied componentwise to a hidden layer is non-expansive if the norm of its output is always at most the norm of its input.  We will add this definition to the next version of the paper.
>
> In Theorem 2.1, the $\log(\beta)$ terms can be removed because they are dominated by $\beta$ terms.
>
> Thank you for your other suggestions — we will make those changes.

---

> > ### Comment · AnonReviewer2 · 2019-11-13
> > **Response to authors' response**
> >
> > Thank you for your response to my review comments. It deepens my understanding.
> >
> >
> > > As we wrote in our response to Reviewer 1, if one just applies a B sqrt(D/N) bound, even if the operator norms of all of the layers are bounded by a constant, the dependence on the bound is exponential in the depth of the network. One contribution of this paper is to recognize that the bounds of Gine and Guillou are especially relevant to deep networks, enabling a linear dependence on the depth.
> >
> > First of all, $O(B \sqrt(D/N))$ in my review comment should have been $O(\sqrt(Dlog B/N))$, the same form as Theorem A.4. Although the main point of my comment does not change, I am sorry if it confused the authors.
> > I agree that the bound the authors derived in section 2.4 improved the dependence on the depth of the network $L$ compared to the bound of Bartlett et al., (2017). However, since the exponential dependence on depth are typical in norm-based bounds (I borrowed this term from Suzuki (2019)), including Bartlett et al. (2017), compression-based bounds [Arora et al., 2019; Suzuki, 2019; Nagarajan and Kolter, 2019] have been proposed and alleviated the dependence on depth. [Wei and Ma, 2019], which Reviewer #3 mentioned, also addressed this problem. I would like to ask authors how they differentiate their work from aformentioned results. Otherwise, I would say that the novelty of removing the exponential dependence on depth is limited.
> >
> > [Arora et al., 2018] S. Arora, R. Ge, B. Neyshabur, and Y.Zhang. Stronger generalization bounds for deepnets via a compression approach. ICML 2018.
> > [Suzuki, 2019] T. Suzuki. Compression based bound for non-compressed network: unified generalization error analysis of large compressible deep neural network. arXiv:1909.11274, 2019.
> > [Nagarajan and Kolter, 2019] V.Nagarajan, and J.Z. Kolter. Deterministic PAC-Bayesian generalization bounds for deep networks via generalizing noise-resilience., ICLR 2019
> > [Wei and Ma, 2019] C. Wei, and T. Ma. Data-dependent Sample Complexity of Deep Neural Networks via Lipschitz Augmentation. arXiv:1905.03684, 2019.
> >
> >
> > > Your point regarding the discussion of Figure 3 is well-taken — the distance from initialization only starts trending down once the networks get pretty big.  We will update our paper to describe this finding in a more refined way.
> >
> > Thank you for considering my comments. I am looking forward to the revised version of the paper.
> >
> >
> > > An activation function that is applied componentwise to a hidden layer is non-expansive if the norm of its output is always at most the norm of its input.  We will add this definition to the next version of the paper.
> > > In Theorem 2.1, the $\log (\beta)$ terms can be removed because they are dominated by $\beta$ terms.
> >
> > Thank you for the explanation. I understand these points.

---

> > > ### Author Response · Authors · 2019-11-13
> > > **Response to AnonReviewer2**
> > >
> > > The paper by Suzuki and the paper by Nagarajan and Kolter were posted on Arxiv after we posted our paper.  The paper by Wei and Ma was also independent of our work.  (It was posted on Arxiv less than three weeks before our paper, uses dissimilar techniques, and has results incomparable in strength with ours.)   All four of the papers use complicated assumptions on the data distribution, whereas our bounds are distribution-free and easily understood.
> > >
> > > As we mentioned in the intro of our paper, the paper by Arora, et al analyzes a compressed variant of the trained network, where we provide bounds for the trained network.
> > >
> > > In Theorem 1.1 of the paper by Wei and Ma, the bound grows cubically in the depth of the network, even if one employs the strong assumption that the largest extent to which any multi-layer subnetwork blows up its input signal is bounded by a constant.  In representative concrete cases discussed in our paper, our bounds are linear in the depth of the network.  Their dependence on the norms of the layers of the networks closely parallels the dependence of Bartlett, et al, so we also achieve a similar improvement on other parameters of the problem over their bounds.  In particular, in the case of all-convolutional networks, their bounds depend on the size of the feature maps, where ours do not.
> > >
> > > We do feel that bounds that capture “wide minima” are also interesting, and we hope that our techniques can be combined with ideas from those papers to obtain stronger results in the case of deep convolutional networks in future work.

---

> > > > ### Comment · AnonReviewer2 · 2019-11-15
> > > > **Thank you for your explanation.**
> > > >
> > > > Thank you for giving us your explanation for the relationship with other works. I will carefully read your comments and check my evaluation again.

---

### Official Review · AnonReviewer1 · 2019-10-25
**Official Blind Review #1**

**Rating:** 3

**Review:**

The paper considers the generalization bound for deep neural networks, specifically, convolutional neural networks, which is one of the popular and crucial topics in the machine learning community, which has gathered a lot of attention.

The paper presents a generalization bound based on the number of parameters, the Lipschitz constant of the loss function and the distance of the final weights from the initialization, without dependence on the dimension of the input. The bound improves upon previous bound in some regimes when the size convolutional kernel is much less than the width of the network, which is a reasonable assumption. The paper also gives another bound which works for fully connected layers with an additional term that is linear with the depth of the network.

The paper has some nice ideas, but the contribution of the paper is not clear for me. The main theorems are based on previous results (Lemma 2.3). And the remaining work of the paper is mainly deriving the Lipchitz bound to be used in the theorem for various kinds of networks. I think this should be clearly stated in the paper.

The experiment part is not quite convincing. It is not clear from the figures that the norm decreases with the number of parameters in the network, which is claimed in the paper.

The writing of the paper also can be improved. The paper presents math, which is nice, but without much intuition explained.

Overall I would not recommend this paper for admission.

**Experience Assessment:**

I do not know much about this area.

**Review Assessment: Checking Correctness Of Derivations And Theory:**

I assessed the sensibility of the derivations and theory.

**Review Assessment: Checking Correctness Of Experiments:**

I assessed the sensibility of the experiments.

**Review Assessment: Thoroughness In Paper Reading:**

I read the paper at least twice and used my best judgement in assessing the paper.

---

> ### Author Response · Authors · 2019-11-07
> **response to AnonReviewer1**
>
> Thank you for your review.
>
> The main contribution of this paper is to capture theoretically the inductive leverage arising from using convolutional layers in deep networks.   We note that a paper treating the two-layer case was published in NeurIPS in 2018.
>
> As far as we know, we are the first authors in the deep learning literature to apply the bounds of Gine and Guillou.  As we highlight in the intro, their bounds appear to be especially useful for deep learning — they are key to enabling the reduction of the dependence in the bounds on the depth of the network from exponential to linear.
>
> Your point regarding the discussion of Figure 3 is well-taken — the distance from initialization only starts trending down once the networks get pretty big.  We will update our paper to describe this finding in a more refined way.

---

### Official Review · AnonReviewer3 · 2019-10-25
**Official Blind Review #3**

**Rating:** 6

**Review:**


The paper describes new norm-based generalization bounds that were specifically adapted to convolutional neural networks. Since convolutional neural networks do not explicitly depend on the input dimension, these bounds share the same property. Further additional improvement over Bartlett et al. ‘17 bound, is that this new bound depends on the sum of the operator norms of the parameter matrices, rather than the product.

The paper is clearly written and self-contained. I appreciate that the authors added a detailed comparison to Bartlett et al. ‘17 bound. However, the main result seems to be very incremental. The experiments are also very limited and not too convincing. Further empirical evaluation is needed to demonstrate progress. I would be willing to increase my score if the authors added a comparison to Wei and Ma ‘19, and more evidence was provided that the bound is tighter for typical convolutional networks found in practice (please see detailed comments below).

Detailed comments:

I see Wei and Ma ‘19 cited in the beginning only, but there is no further comparison. They also proved bounds with similar dependencies. How do the bounds presented in the paper compare to Wei and Ma bounds?

What is the dependence of the constant C on \eta in the bounds presented in Theorem 2.1? It is unclear what trade-off comes with eta and how the empirical risk term is balanced with the complexity term, since \eta only appears next to the empirical risk term.

The authors demonstrate via a concrete example that there exists a setting (depending on epsilon), under which this new bound (up to constants) is tighter than Bartlett et al. bound. Three things remain unclear to me:
 - How do the constants differ? Is the bound presented in the paper tighter in absolute terms?
 - Is the bound tighter when the norms in the bounded are measured on typical trained neural network weights? An analysis of a few networks used in practice would make the comparison more meaningful (included the comparison to Wei and Ma).
 - Is the bound not worse than a VC bound in any (reasonable) setting? If not, is the bound tighter under typical settings when training standard vision networks?

Other minor comments. In the introduction, the authors:
 - say that their bounds are size-free, which refers to the bounds not having an explicit dependence on the input size. In my opinion, this comes almost “for free” when using convolutional neural network. Also, I think that size-free in the title is misleading, and should be replaced with input size-free.
 - mention that most recent bounds depend on the distance from the initialization instead of the size of the weights. This idea was first presented in Dziugate and Roy ‘17, which does not seem to be cited there.


*** UPDATE ***

I've reread the rebuttals and feel that most of my concerns have been addressed. I increased my score to weak accept.


**Experience Assessment:**

I have published in this field for several years.

**Review Assessment: Checking Correctness Of Derivations And Theory:**

I assessed the sensibility of the derivations and theory.

**Review Assessment: Checking Correctness Of Experiments:**

I assessed the sensibility of the experiments.

**Review Assessment: Thoroughness In Paper Reading:**

I read the paper thoroughly.

---

> ### Author Response · Authors · 2019-11-07
> **response to AnonReviewer3**
>
> Thank you for your review.
>
> Overall, our results are incomparable in strength with the bounds in Wei and Ma’s paper, which was posted on Arxiv a few weeks before ours.  It is likely that the ideas from our analysis can be combined with theirs to strengthen both.  Because their bounds are data-dependent, a comparison in the style of the comparison that we made with the BFT bounds is not possible.  The WM bounds are in terms of the operator norms of interlayer Jacobians;  these govern how much any subnetwork can increase the length of its input.   Even if the operator norms of all layers are bounded by a constant, these interlayer Jacobians can grow exponentially with the depth of the network, whereas our bounds, even in the fully connected case, grow linearly with the depth of the network.  On the other hand, a main point of their paper is that this exponential growth may not be seen in natural data.  However, note that, in Theorem 1.1 of their paper, the bound grows cubically in the depth of the network, even if one employs the strong assumption that the largest operator norm of any interlayer Jacobian is a constant.  In representative concrete cases discussed in our paper, our bounds are linear in the depth of the network.  Their dependence on the norms of the layers of the networks closely parallels the dependence of Bartlett, et al, so we also achieve a similar improvement on other parameters of the problem over their bounds.  In particular, in the case of all-convolutional networks, their bounds depend on the size of the feature maps, where ours do not.
>
> C depends on $\eta$ roughly as $1/\eta^2$.  However, the first bound is meant to be applied with a value of eta that is not too small.  Note that if the Bayes error rate is zero (the “realizable” case) the choice of eta does not matter at all, and if the Bayes error rate is small, then twice the Bayes error rate is also small.  If the Bayes error rate is large, then the second bound is more relevant.
>
> We do not know of a previously published bound on the VC-dimension of deep convolutional networks that is independent of the size of intermediate feature maps.  Because the networks treated in our analysis have multiple outputs, and because the main consequence of our bounds is a margin bound, generalizations of the VC-dimension like fat-shattering would potentially be more relevant.  We strongly suspect that a fat-shattering bound is implicit in our analysis.  However, we note that, at least since Zhang’s 2002 JMLR paper, researchers have recognized that stronger generalization bounds with simpler proofs can be obtained by directly bounding covering numbers as we do.
>
> We did reference Dziugate and Roy, and, in the intro, we also mentioned “previous bounds in terms of distance to initialization”, acknowledging that this idea is not a contribution of this paper.  In the next version of our paper, however, we will more clearly indicate that Dziugate and Roy were the first to prove a bound like this.

---

> > ### Comment · AnonReviewer3 · 2019-11-12
> > **size-free**
> >
> > SIZE-FREE
> >
> > First, I would like to make sure I understand exactly what the authors mean by the "size of the feature maps". Is it the sizes of input/intermediate layers? (different from the kernel size). If so, while Bartlett et al. bounds do depend on the largest width, Wei and Ma's bounds are independent of the width of the network.
> >
> > Otherwise, the response above would suggest that the bounds presented in the paper are independent of the size of the kernels. However, the bounds presented in Theorem 2.1 depend on W, which depends on the number of parameters. I would be grateful if the authors could clarify.
> >
> > COMPARISON to Wei and MA and EMPIRICAL INVESTIGATION
> >
> > It looks like authors missed my questions regarding empirically investigating the quantities appearing in their bounds and in related work bounds. I would be grateful if the authors could respond (copied below). My final evaluation rests on further empirical evidence of the relevance of the bounds.
> >
> > (the rest is copied from the original review):
> > The authors demonstrate via a concrete example that there exists a setting (depending on epsilon), under which this new bound (up to constants) is tighter than Bartlett et al. bound. Three things remain unclear to me:
> >  - How do the constants differ? Is the bound presented in the paper tighter in absolute terms?
> >  - Is the bound tighter when the norms in the bounded are measured on typical trained neural network weights? An analysis of a few networks used in practice would make the comparison more meaningful (included the comparison to Wei and Ma).

---

> > > ### Author Response · Authors · 2019-11-12
> > > **response to AnonReviewer3**
> > >
> > > Thank you for reading our response, and your second round of questions.
> > >
> > > You are correct about what we mean by the size of the intermediate feature maps.
> > >
> > > For analysis of convolutional networks, Theorem 1.1 of Wei and Ma’s paper may only be applied using a representation as a fully connected network.  If the feature map is $d \times d$, each entry of the kernel is repeated $d^2$ times in this representation of its layer.  Thus, the mixed norms in WM’s Theorem 1.1 do grow with $d$.
> > >
> > > We have not derived specific constants for our bounds.

---

### Decision · Program_Chairs · 2019-12-19

**Decision:**

Accept (Poster)

**Comment:**

The authors present several theorems bounding the generalization error of a class of conv nets (CNNs) with high probability by

      O(sqrt(W(beta + log(lambda)) + log(1/delta)]/sqrt(n)),

where W is the number of weights, beta is the distance from initialization in operator norm, lambda is the margin, n is the number of data, and the bound holds with prob. at least 1-delta. (They also present a bound that is tighter when the empirical risk is small.)

The bounds are "size free" in the sense that they do not depend on the size of the *input*, which is assumed to be, say, a d x d image. While there is dependence on the number of parameters, W, there is no implicit dependence on d here.

The paper received the following feedback:

1. Reviewer 3 mostly had clarifying questions, especially with respect to (essentially independent) work by Wei and Ma. Reviewer 3 also pressed the authors to discuss how the bounds compared in absolute terms to the bounds of Bartlett et al. The authors stated that they did not have explicit constants to make such a comparison. Reviewer 3 was satisfied enough to raise their score to a 6.

2. Reviewer 1 admitted they were not experts and raised some issues around novelty/simplicity. I do not think the simplicity of the paper is a drawback. The reviewers unfortunately did not participate in the rebuttal, despite repeated attempts.

3. Reviewer 2 argued for weak reject, despite an interaction with the authors. The reviewer raised the issue of bounds based on control of the Lipschitz constant. The conversation was slightly marred by a typo in the reviewers original comment. I don't believe the authors ultimately responded to the reviewer's point. There was another discussion about simultaneously work and compression-based bounds. I would agree with the authors that they need not have cited simultaneous work, especially since the details are quite different. Ultimately, this reviewer still argued for rejection (weakly).

After the rebuttal period ended, the reviewers raised some further concerns with me. I tried to assess these on my own, and ended up with my own questions.

I raise these in no particular order. Each of them may have a simple resolution. In that case, the authors should take them as possible sources of confusion. Addressing them may significantly improve the readability of the paper.

i. Lemma A.3. The order of quantification is poorly expressed and so I was not confident in the statement. In particular, the theorem starts \forall \eta >0 \exists C, .... but then C is REINTRODUCED later, subsequent to existential quantification over M, B, and d and so it seems there is dependence. If there is no dependence, this presentation is sloppy and should be fixed.

ii. Lemma A.4, the same dependence of C on M, B and d holds here and this is quite problematic for the later applications. If this constant is independent of these quantities, then the order of quantifiers has been stated incorrectly. Again, this is sloppy if it is wrong. If it's correct, then we need to know how C grows.

Based on other claims by the authors, it is my understanding that, in both cases, the constant C does not depend on M, B, or d. Regardless, the authors should clarify the dependence. If C does in fact depend on these quantities, and the conclusions change, the paper should be retracted.

iii. Proof of Lemma 2.3. I'd remind the reader that the parametrization maps the unit ball to G.

iv. The bound depends on control of operator norms and empirical margins. It is not clear how these interact and whether, for margin parameters necessary to achieve small empirical margin risk, the bounds pick up dependence on other aspects of the learning problem (e.g., depth). I think the only way to assess this would be to investigate these quantities empirically, say, by varying the size and depth of the network on a fixed data set, trained to achieve the same empirical risk (or margin).

I'll add that I was also disappointed that the authors did not attempt to address any of the issues by a revision of the actual paper. In particular, the authors promise several changes that would have been straightforward to make in the two weeks of rebuttal. Instead, the reviewers and myself are left to imagine how things would change. I see at least two promises:

A. To walk back some of the empirical claims about distance from initialization that are based on somewhat flimsy empirical evaluations. I would add to this the need to investigate how the margin and operator norms depend on depth empirically.

B. Attribute Dziugate and Roy for establishing the first bounds in terms of distance from initialization, though their bounds were numerical. I think a mention of simultaneously work would also be generous, even if not strictly necessary.